# FOUR THINGS EVERYONE SHOULD KNOW TO IMPROVE BATCH NORMALIZATION

**Cecilia Summers**
Department of Computer Science
University of Auckland
`cecilia.summers.07@gmail.com`

**Michael J. Dinneen**
Department of Computer Science
University of Auckland
`mjd@cs.auckland.ac.nz`

## ABSTRACT

A key component of most neural network architectures is the use of normalization layers, such as Batch Normalization. Despite its common use and large utility in optimizing deep architectures, it has been challenging both to generically improve upon Batch Normalization and to understand the circumstances that lend themselves to other enhancements. In this paper, we identify four improvements to the generic form of Batch Normalization and the circumstances under which they work, yielding performance gains across all batch sizes while requiring no additional computation during training. These contributions include proposing a method for reasoning about the current example in inference normalization statistics, fixing a training vs. inference discrepancy; recognizing and validating the powerful regularization effect of Ghost Batch Normalization for small and medium batch sizes; examining the effect of weight decay regularization on the scaling and shifting parameters $\gamma$ and $\beta$; and identifying a new normalization algorithm for very small batch sizes by combining the strengths of Batch and Group Normalization. We validate our results empirically on six datasets: CIFAR-100, SVHN, Caltech-256, Oxford Flowers-102, CUB-2011, and ImageNet.

## 1 INTRODUCTION

Neural networks have transformed machine learning, forming the backbone of models for tasks in computer vision, natural language processing, and robotics, among many other domains (Krizhevsky & Hinton, 2009; He et al., 2017; Levine et al., 2016; Sutskever et al., 2014; Graves et al., 2013). A key component of many neural networks is the use of normalization layers such as Batch Normalization (Ioffe & Szegedy, 2015), Group Normalization (Wu & He, 2018), or Layer Normalization (Ba et al., 2016), with Batch Normalization the most commonly used for vision-based tasks. While the true reason why these methods work is still an active area of research (Santurkar et al., 2018), normalization techniques typically serve the purpose of making neural networks more amenable to optimization, allowing the training of very deep networks without the use of careful initialization schemes (Simonyan & Zisserman, 2015; Zhang et al., 2019), custom nonlinearities (Klambauer et al., 2017), or other more complicated techniques (Xiao et al., 2018). Even in situations where training without normalization layers is possible, their usage can still aid generalization (Zhang et al., 2019). In short, normalization layers make neural networks train faster and generalize better.

Despite this, it has been challenging to improve normalization layers. In the general case, a new approach would need to be uniformly better than existing normalization methods, which has proven difficult. It has even been difficult to tackle a simpler task: characterizing when specific changes to common normalization approaches might yield benefits. In all, this has created an environment where approaches such as Batch Normalization are still used as-is, unchanged since their creation.

In this work we identify four techniques that everyone should know to improve their usage of Batch Normalization, arguably the most common method for normalization in neural networks. Taken together, these techniques apply in all circumstances in which Batch Normalization is currently used, ranging from large to very small batch sizes, including one method which is even useful when the batch size $B = 1$, and for each technique we identify the circumstances under which it is expected to be of use. In summary, our contributions are:

1. A way to more effectively use the current example during inference, fixing a discrepancy between training and inference that had been previously overlooked,

2. Identifying Ghost Batch Normalization, a technique designed for very large-batch multi-GPU training (Hoffer et al., 2017), as surprisingly effective even in the medium-batch, single-GPU regime,

3. Recognizing weight decay of the scaling and centering variables $\gamma$ and $\beta$ as a valuable source of regularization, an unstudied detail typically neglected, and

4. Proposing a generalization of Batch and Group Normalization in the small-batch setting, effectively making use of cross-example information present in the minibatch even when such information is not enough for effective normalization on its own.

Experimentally, we study the most common use-case of Batch Normalization, image classification, which is fundamental to most visual problems in machine learning. In total, these four techniques can have a surprisingly large effect, improving accuracy by over 6% on one of our benchmark datasets while only changing the usage of Batch Normalization layers.

We have released code at `https://github.com/ceciliaresearch/four_things_batch_norm`.

## 2 RELATED WORK/BACKGROUND ON NORMALIZATION METHODS

Most normalization approaches in neural networks, including Batch Normalization, have the general form of normalizing their inputs $x_i$ to have a learnable mean and standard deviation:

$$\hat{x}_i = \gamma \frac{x_i - \mu_i}{\sqrt{\sigma_i^2 + \epsilon}} + \beta \tag{1}$$

where $\gamma$ and $\beta$ are the learnable parameters, typically initialized to 1 and 0, respectively. Where approaches typically differ is in how the mean $\mu_i$ and variance $\sigma_i^2$ are calculated.

Batch Normalization (Ioffe & Szegedy, 2015), the pioneering work in normalization layers, defined $\mu_i$ and $\sigma_i^2$ to be calculated for each channel or feature map separately across a minibatch of data. For example, in a convolutional layer, the mean and variance are computed across all spatial locations and training examples in a minibatch. During inference, these statistics are replaced with an exponential moving average of the mean and variance, making inference behavior independent of inference batch statistics. The effectiveness of Batch Normalization is undeniable, playing a key role in nearly all state-of-the-art convolutional neural networks since its discovery (Szegedy et al., 2016; 2017; He et al., 2016a;b; Zoph & Le, 2017; Zoph et al., 2018; Hu et al., 2018; Howard et al., 2017; Sandler et al., 2018). Despite this, there is still a fairly limited understanding of Batch Normalization's efficacy — while Batch Normalization's original motivation was to reduce internal covariate shift during training (Ioffe & Szegedy, 2015), recent work has instead proposed that its true effectiveness stems from making the optimization landscape smoother (Santurkar et al., 2018).

One weakness of Batch Normalization is its critical dependence on having a reasonably large batch size, due to the inherent approximation of estimating the mean and variance with a single batch of data. Several works propose methods without this limitation: Layer Normalization (Ba et al., 2016), which has found use in many natural language processing tasks (Vaswani et al., 2017), tackles this by calculating $\mu_i$ and $\sigma_i^2$ over all channels, rather than normalizing each channel independently, but does not calculate statistics across examples in each batch. Instance Normalization (Ulyanov et al., 2016), in contrast, only calculates $\mu_i$ and $\sigma_i^2$ using the information present in each channel, relying on the content of each channel at different spatial locations to provide effective normalization statistics. Group Normalization (Wu & He, 2018) generalizes Layer and Instance Normalization, calculating statistics in "groups" of channels, allowing for stronger normalization power than Instance Normalization, but still allowing for each channel to contribute significantly to the statistics used for its own normalization. The number of normalization groups per normalization layer is typically set to a global constant in group normalization, though alternatives such as specifying the number of channels per group have also been tried (Wu & He, 2018).

Besides these most common approaches, many other forms of normalization also exist: Weight Normalization (Salimans & Kingma, 2016) normalizes the weights of each layer instead of the inputs,

parameterizing them in terms of a vector giving the direction of the weights and an explicit scale, which must be initialized very carefully. Decorrelated Batch Normalization (Huang et al., 2018) performs ZCA whitening in its normalization layer, and Iterative Normalization (Huang et al., 2019) makes it more efficient via a Newton iteration approach. Cho & Lee (2017) analyze the weights in Batch Normalization from the perspective of a Riemannian manifold, yielding new optimization and regularization methods that utilize the manifold's geometry.

Targeting the small batch problem, Batch Renormalization (Ioffe, 2017) uses the moving average of batch statistics to normalize during training, parameterized in such a way that gradients still propagate through the minibatch mean and standard deviation, but introduces two new hyperparameters and still suffers somewhat diminished performance in the small-batch setting. Guo et al. (2018) tackle the small batch setting by aggregating normalization statistics over multiple forward passes.

Recently, Switchable Normalization (Luo et al., 2019) aims to learn a more effective normalizer by calculating $\mu_i$ and $\sigma_i^2$ as learned weighted combinations of the statistics computed from other normalization methods. While flexible, care must be taken for two reasons: First, as the parameters are learned differentiably, they are fundamentally aimed at minimizing the training loss, rather than improved generalization, which typical hyperparameters are optimized for on validation sets. Second, the choice of which normalizers to include in the weighted combination remains important, manifesting in Switchable Normalization's somewhat worse performance than Group Normalization for small batch sizes. Differentiable Dynamic Normalization (Luo et al., 2019) fixes the latter point, learning an even more flexible normalization layer. Beyond these, there are many approaches we omit for lack of space (Littwin & Wolf, 2018; Deecke et al., 2019; Hoffer et al., 2018; Klambauer et al., 2017; Xiao et al., 2018; Zhang et al., 2019).

## 3   IMPROVING NORMALIZATION: WHAT EVERYONE SHOULD KNOW

In this section we detail four methods for improving Batch Normalization. We also refer readers to the Appendix for a discussion of methods which do *not* improve normalization layers (sometimes surprisingly so). For clarity, we choose to interleave descriptions of the methods with experimental results, which aids in understanding each of the approaches as they are presented. We experiment with four standard image-centric datasets in this section: CIFAR-100, SVHN, Caltech-256, and ImageNet, and report results on validation datasets in order to fully describe each approach without contaminating test-set results. We give results on test sets, and experimental details in Sec. 4.

### 3.1   INFERENCE EXAMPLE WEIGHING

Batch Normalization has a disparity in function between training and inference: As previously noted, Batch Normalization calculates its normalization statistics over each minibatch of data separately while training, but during inference a moving average of training statistics is used, simulating the expected value of the normalization statistics. Resolving this disparity is a common theme among methods that have sought to replace Batch Normalization (Ba et al., 2016; Ulyanov et al., 2016; Salimans & Kingma, 2016; Wu & He, 2018; Ioffe, 2017). Here we identify a key component of this training versus inference disparity which can be fixed within the context of Batch Normalization itself, improving it in the general case: when using a moving average during inference, each example does not contribute to its own normalization statistics.

To give an example of the effect this has, we consider the output range of Batch Normalization. During training, due to the inclusion of each example in its own normalization statistics, it can be shown[1] that the minimum possible output of a Batch Normalization layer is:

$$\min_{x_0,\dots,x_{B-1}} \gamma \frac{x_0 - \mu_i}{\sqrt{\sigma_i^2 + \epsilon}} + \beta = -\gamma\sqrt{B-1} + \beta \tag{2}$$

with a corresponding maximum value of $\gamma\sqrt{B-1}+\beta$, where $B$ is the batch size, and we assume for simplicity that Batch Norm is being applied non-convolutionally. In contrast, during inference the output range of Batch Normalization is *unbounded*, creating a discrepancy. Morever, this actually happens for real networks: the output range of a network with Batch Normalization is wider during inference than during training (see Sec. B in Appendix).

---

[1]Proof in Appendix Sec. A

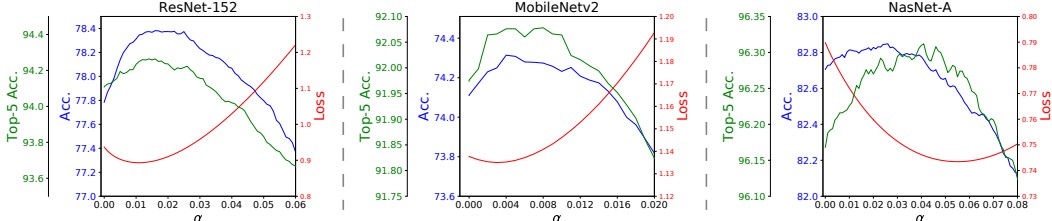

Figure 1: Effect of the example-weighing hyperparameter $\alpha$ on ImageNet for ResNet-152, MobileNetV2, and NASNet-A, measuring top-1 and top-5 accuracies and the cross-entropy loss.

Fortunately, once this problem has been realized, it is possible to fix — we need only figure out how to incorporate example statistics during inference. Denoting $m_x$ as the moving average over $x$ and $m_{x^2}$ the corresponding moving average over $x^2$, we apply the following normalization:

$$
\begin{aligned}
\mu_i &= \alpha E[x_i] + (1 - \alpha)m_x \\
\sigma_i^2 &= (\alpha E[x_i^2] + (1 - \alpha)m_{x^2}) - \mu_i^2 \\
\hat{x}_i &= \gamma \frac{x_i - \mu_i}{\sqrt{\sigma_i^2 + \epsilon}} + \beta
\end{aligned}
\tag{3}
$$

where $\alpha$ is the contribution of $x_i$ to the normalization statistics, and we have reparameterized the variance as $\sigma_i^2 = E[x_i^2] - E[x_i]^2$.

Given this formulation, a natural question is the choice of the parameter $\alpha$, where $\alpha = 0$ corresponds to the classical inference setting of Batch Normalization and $\alpha = 1$ replicates the setting of techniques which do not use cross-image information in calculating normalization statistics. Intuitively, it would make sense for the optimal value to be $\alpha = \frac{1}{B}$. However, this turns out to not be the case — instead, $\alpha$ is a hyperparameter best optimized on a validation set, whose optimal value may depend the model, dataset, and metric being optimized. While counterintuitive, this can be explained by the remaining set of differences between training and inference: for a basic yet fundamental example, the fact that the model has been fit on the training set (also typically with data augmentation) may produce systematically different normalization statistics between training and inference.

An advantage of this technique is that we can apply it retroactively to any model trained with Batch Normalization, allowing us to verify its efficacy on a wide variety of models. In Fig. 1 we show the effect of $\alpha$ on the ImageNet ILSVRC 2012 validation set (Russakovsky et al., 2015) for three diverse models: ResNet-152 (He et al., 2016b), MobileNetV2 (Sandler et al., 2018), and NASNet-A Large (Zoph et al., 2018)[2]. On ResNet-152, for example, proper setting of $\alpha$ can increase accuracy by up to 0.6%, top-5 accuracy by 0.16%, and loss by a relative 4.7%, which are all quite significant given the simplicity of the approach, the competitiveness of ImageNet as a benchmark, and the fact that the improvement is essentially "free" — it involves only modifying the inference behavior of Batch Normalization layers, and does not require any re-training. Across models, the optimal value for $\alpha$ was largest for NASNet-A, the most memory-intensive (and therefore smallest batch size) model of the three. We refer the reader to the Appendix for additional plots with larger ranges of $\alpha$.

Surprisingly, it turns out that this approach can have positive effects on models trained without any cross-image normalization at all, such as models trained with Group Normalization (Wu & He, 2018). We demonstrate this in Fig. 2, where we find that adding a tiny amount of information from the moving average statistics can actually result in small improvements, with relatively larger improvements in accuracy on Caltech-256 and cross entropy loss on CIFAR-100 and SVHN. This finding is extremely surprising, since adding in any information from the moving averages at all represents a clear difference from the training setting of Group Normalization. Similar to the unintuitive optimal value for $\alpha$, we hypothesize that this effect is due to other differences in the settings of training and inference: for example, models are generally trained on images with the application of data augmentation, such as random cropping. During inference, though, images appear unperturbed, and it might be the case that incorporating information from the moving averages is a way of influencing the model's intermediate activations to be more similar to those of data augmented

---

[2]Models obtained from (Silberman & Guadarrama).

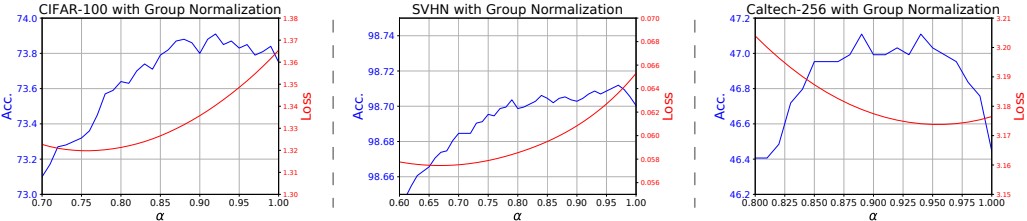

Figure 2: Effect of the example-weighing hyperparameter $\alpha$ for models trained with Group Normalization on CIFAR-100, SVHN, and Caltech-256.

images, which it has been trained on. This mysterious behavior may also point to more general approaches for resolving training-inference discrepancies, and is worthy of further study.

Last, we also note very recent work (Singh & Shrivastava, 2019) which examines a similar approach for incorporating the statistics of an example during inference time, using per-layer weights and optimizing with a more involved procedure that encourages similar outputs to the training distribution.

**Summary:** Inference example weighing resolves one disparity between training and inference for Batch Normalization, is uniformly beneficial across all models and very easy to tune to metrics of interest, and can be used with any model trained with Batch Normalization, even retroactively.

### 3.2 GHOST BATCH NORMALIZATION FOR MEDIUM BATCH SIZES

Ghost Batch Normalization, a technique originally developed for training with very large batch sizes across many accelerators (Hoffer et al., 2017), consists of calculating normalization statistics on disjoint subsets of each training batch. Concretely, with an overall batch size of $B$ and a "ghost" batch size of $B'$ such that $B'$ evenly divides $B$, the normalization statistics for example $i$ are calculated as

$$
\mu_i = \frac{1}{B'} \sum_{j=1}^{B} x_j \left[ \left\lfloor \frac{jB'}{B} \right\rfloor = \left\lfloor \frac{iB'}{B} \right\rfloor \right]
$$

$$
\sigma_i^2 = \frac{1}{B'} \sum_{j=1}^{B} x_j^2 \left[ \left\lfloor \frac{jB'}{B} \right\rfloor = \left\lfloor \frac{iB'}{B} \right\rfloor \right] - \mu_i^2
$$

(4)

where $[\cdot]$ is the Iverson bracket, with value 1 if its argument is true and 0 otherwise. Ghost Batch Normalization was previously found to be an important factor in reducing the generalization gap between large-batch and small-batch models (Hoffer et al., 2017), and has since been used by subsequent research rigorously studying the large-batch regime (Shallue et al., 2018). Here, we show that it can also be useful in the medium-batch setting[3].

Why might Ghost Batch Normalization be useful? One reason is its power as a regularizer: due to the stochasticity in normalization statistics caused by the random selection of minibatches during training, Batch Normalization causes the representation of a training example to randomly change every time it appears in a different batch of data. Ghost Batch Normalization, by decreasing the number of examples that the normalization statistics are calculated over, increases the strength of this stochasticity, thereby increasing the amount of regularization. Based on this hypothesis, we would expect to see a unimodal effect of the Ghost Batch Normalization size $B'$ on model performance — a large value of $B'$ would offer somewhat diminished performance as a weaker regularizer, a very low value of $B'$ would have excess regularization and lead to poor performance, and an intermediate value would offer the best tradeoff of regularization strength.

We confirm this intuition in Fig. 3. Surprisingly, just using this one simple technique was capable of improving performance by 5.8% on Caltech-256 and 0.84% on CIFAR-100, which is remarkable given it has no additional cost during training. On SVHN, though, where baseline performance is already a very high 98.79% and models do not overfit much, usage of Ghost Batch Normalization

---

[3]We experiment with batch sizes up to 128 in this work.

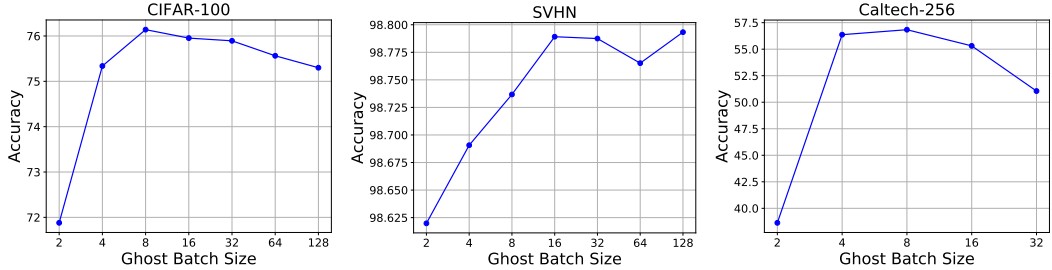

Figure 3: Accuracy vs. Ghost Batch Normalization size for CIFAR-100, SVHN, and Caltech-256.

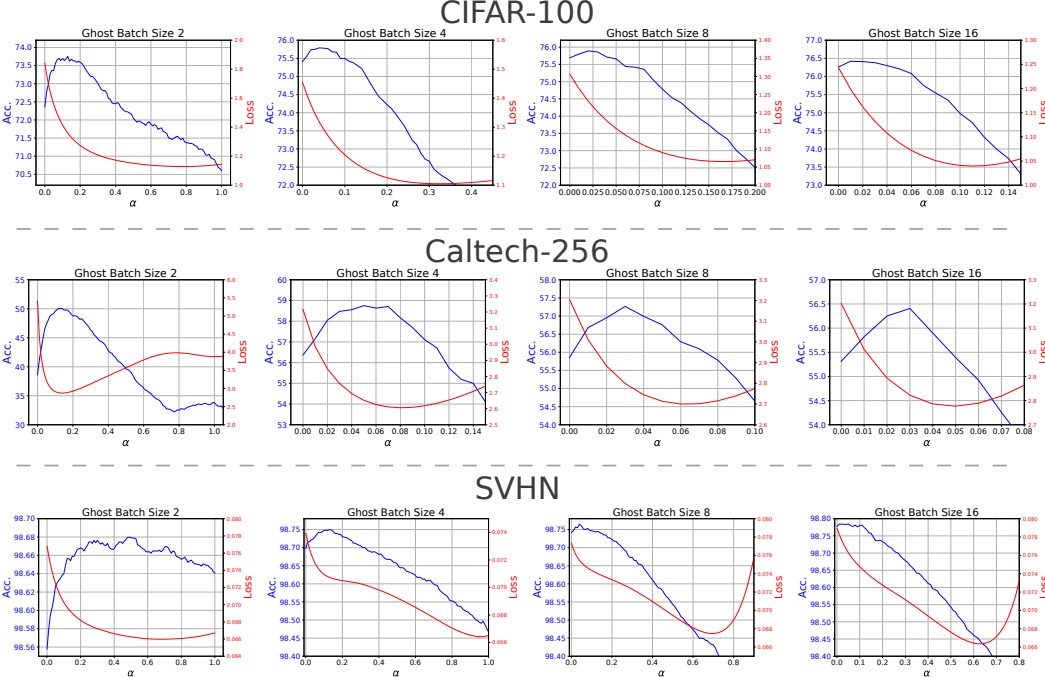

Figure 4: The complementary effects of Inference Example Weighing (Sec. 3.1) and Ghost Batch Normalization (Sec. 3.2) on CIFAR-100, SVHN, and Caltech-256.

did not result in an improvement, giving evidence that at least part of its effect is regularization in nature. In practice, $B'$ may be treated as an additional hyperparameter to optimize.

As a bonus, Ghost Batch Normalization has a synergistic effect with inference example weighing — it has the effect of making each example more important in calculating its own normalization statistics $\mu_i$ and $\sigma_i^2$, with greater effect the smaller $B'$ is, precisely the setting that inference example weighing corrects for. We show these results in Fig. 4, where we find increasing gain from inference example weighing as $B'$ is made smaller, a gain that compounds from the benefits of Ghost Batch Normalization itself. Interestingly, these examples also demonstrate that accuracy and cross-entropy, the most commonly-used classification loss, are only partially correlated, with the optimal values for the inference example weight $\alpha$ sometimes differing wildly between the two (*e.g.* for SVHN).

**Summary:** Ghost Batch Normalization is beneficial for all but the smallest of batch sizes, has no computational overhead, is straightforward to tune, and can be used in combination with inference example weighing to great effect.

### 3.3 BATCH NORMALIZATION AND WEIGHT DECAY

Weight decay (Krogh & Hertz, 1992) is a regularization technique that scales the weight of a neural network after each update step by a factor of $1 - \delta$, and has a complex interaction with Batch Normalization. At first, it may even seem paradoxical that weight decay has any effect in a network trained with Batch Normalization, as scaling the weights immediately before a normalization layer by any non-zero constant has mathematically almost no effect on the output of the normalization layer (and no effect at all when $\epsilon = 0$). However, weight decay actually has a subtle effect on the effective learning rate of networks trained with Batch Normalization — without weight decay, the weights in a batch-normalized network grow to have large magnitudes, which has an inverse effect on the effective learning rate, hampering training (Hoffer et al., 2018; van Laarhoven, 2017).

Here we turn our attention to the less studied scale and bias parameters common in most normalization methods, $\gamma$ and $\beta$. As far as we are aware, the effect of regularization on $\gamma$ and $\beta$ has not been studied to any great extent — Wu & He (2018) briefly mention weight decay with these parameters, where weight decay was used when training from scratch, but not fine-tuning, two other papers (Goyal et al., 2017; He et al., 2016a) have this form of weight decay explicitly turned off, and He et al. (2019) encourage disabling weight decay on $\gamma$ and $\beta$, but ultimately find diminished performance by doing so.

Unlike weight decay on weights in *e.g.* convolutional layers, which typically directly precede normalization layers, weight decay on $\gamma$ and $\beta$ can have a regularization effect so long as there is a path in the network between the layer in question and the ultimate output of the network, as if such paths do not pass through another normalization layer, then the weight decay is never "undone" by normalization. This structure is only common in certain types of architectures; for example, Residual Networks (He et al., 2016a;b) have such paths for many of their normalization layers due to the chaining of skip-connections. However, Inception-style networks (Szegedy et al., 2016; 2017) have no residual connections, and despite the fact that each "Inception block" branches into multiple paths, every Batch Normalization layer other than those in the very last block do not have a direct path to the network's output.

We evaluated the effects of weight decay on $\gamma$ and $\beta$ on CIFAR-100 across 10 runs, where we found that incorporating it improved accuracy by a small but significant $0.3\%$ ($P = 0.002$). Interestingly, even though $\gamma$ has a multiplicative effect, we did not find it mattered whether $\gamma$ was regularized to 0 or 1 ($P = 0.46$) — what was important was whether it had weight decay applied at all.

We did the same comparison on Caltech-256 with Inception-v3 and ResNet-50 networks, where we found evidence that the network architecture plays a crucial effect: for Inception-v3, incorporating weight decay on $\gamma$ and $\beta$ actually *hurt* performance by $0.13\%$ (mean across 3 trials), while it improved performance for the ResNet-50 network by $0.91\%$, supporting the hypothesis that the structure of paths between layers and the network's output are what matter in determining its utility.

On SVHN, where the baseline ResNet-18 already had a performance of $98.79\%$, we found a similar pattern as with Ghost Batch Normalization — introducing this regularization produced no change.

**Summary:** Regularization in the form of weight decay on the normalization parameters $\gamma$ and $\beta$ can be applied to any normalization layer, but is only effective in architectures with particular connectivity properties like ResNets and in tasks for which models are already overfitting.

### 3.4 GENERALIZING BATCH AND GROUP NORMALIZATION FOR SMALL BATCHES

While Batch Normalization is very effective in the medium to large-batch setting, it still suffers when not enough examples are available to calculate reliable normalization statistics. Although we have shown that techniques such as Inference Example Weighing (Sec. 3.1) can help significantly with this, it is still only a partial solution. At the same time, Group Normalization (Wu & He, 2018) was designed for a batch size of $B = 1$ or greater, but ignores all cross-image information.

In order to generalize Batch and Group Normalization in the batch size $B > 1$ case, we propose to expand the grouping mechanism of Group Normalization from being over only channels to being over both channels and examples — that is, normalization statistics are calculated both *within* groups of channels of each example and *across* examples in groups within each batch [4].

---

[4]See submitted code for specific implementation details.

In principle, this would appear to introduce an additional hyperparameter on top of the number of channel groups used by Group Normalization, both of which would need to be optimized by expensive end-to-end runs of model training. However, in this case we can actually take advantage of the fact that the target batch size is small: if the batch size $B$ is ever large enough that having multiple groups in the example dimension is useful, then it is *also* large enough to eschew usage of the channel groups from Group Normalization, in a regime where either vanilla Batch Normalization or Ghost Batch Normalization is more effective. Thus, when dealing with a small batch size, in practice we only need to optimize over the same set of hyperparameters as Group Normalization.

To demonstrate, we target the extreme setting of $B = 2$, and incorporate Inference Example Weighing to all approaches. For CIFAR-100, this approach improves validation set performance over a tuned Group Normalization by $0.69\%$ in top-1 accuracy (from $73.91\%$ to $74.60\%$, average over three runs), and on Caltech-256, performance dramatically improved by $5.0\%$ (from $48.2\%$ to $53.2\%$, average over two runs). However, this approach has one downside: due to differences in feature statistics across examples, when using only two examples the variability in the normalization statistics can still be quite high, even when using multiple channels within each normalization group. As a result, a regularization effect can occur, which may be undesirable for tasks which models are not overfitting much. As in Sec. 3.2 and Sec. 3.3, we see this effect in SVHN, where this approach is actually ever so slightly worse than Group Normalization on the validation set (from $98.75\%$ to $98.73\%$). On such datasets and tasks, it may be more fruitful to invest in higher-capacity models.

**Summary:** Combining Group and Batch Normalization leads to more accurate models in the setting of batch sizes $B > 1$, and can have a regularization effect due to Batch Normalization's variability in statistics when calculated over small batch sizes.

## 4 ADDITIONAL EXPERIMENTS

### 4.1 EXPERIMENTAL DETAILS

All results in Sec. 3 were performed on the validation datasets of each respective dataset (this section examines test set performance after hyperparameters have been optimized). Of the six datasets we experiment with, only ImageNet (Russakovsky et al., 2015) and Flowers-102 (Nilsback & Zisserman, 2008) have their own pre-defined validation split, so we constructed validation splits for the other datasets as follows: for CIFAR-100 (Krizhevsky & Hinton, 2009), we randomly took 40,000 of the 50,000 training images for the training split, and the remaining 10,000 as a validation split. For SVHN (Netzer et al., 2011), we similarly split the 604,388 non-test images in a 80-20% split for training and validation. For Caltech-256, no canonical splits of any form are defined, so we used 40 images of each of the 256 categories for training, 10 images for validation, and 30 for testing. For CUB-2011, we used 25% of the given training data as a validation set.

The model used for CIFAR-100 and SVHN was ResNet-18 (He et al., 2016b;a) with 64, 128, 256, and 512 filters across blocks. For Caltech-256, a much larger Inception-v3 (Szegedy et al., 2016) model was used, and we additionally experiment with ResNet-152 (He et al., 2016b) on Flowers-102 and CUB-2011 in Sec. 4.3. All experiments were done on two Nvidia Geforce GTX 1080 Ti GPUs.

### 4.2 COMBINING ALL FOUR: IMPROVEMENTS ACROSS BATCH SIZES

Here we show the end-to-end effect of these four improvements on the test sets of each dataset, comparing against both Batch and Group Normalization, with a batch size $B = 128$. We plot results for CIFAR-100 and Caltech-256 in Fig. 5 (a), comparing against Group Normalization and an idealized Batch Normalization with constant performance across batch sizes (simulating if the problematic dependence of Batch Norm on the batch size were completely solved). On CIFAR-100, we see improvements against the best available baseline across all batch sizes.

For medium to large batch sizes ($B \geq 4$), improvements are driven by the combination of Ghost Batch Normalization (Sec. 3.2), Inference Example Weighing (Sec. 3.1), and weight decay introduced on $\gamma$ and $\beta$ (Sec. 3.3). To aid in distinguishing between these effects, we also plot the impact of Ghost Batch Normalization alone, which we find particularly impactful as long as long as the batch size isn't too small ($B > 2$). Turning to very small batch sizes, for $B = 1$ improvements

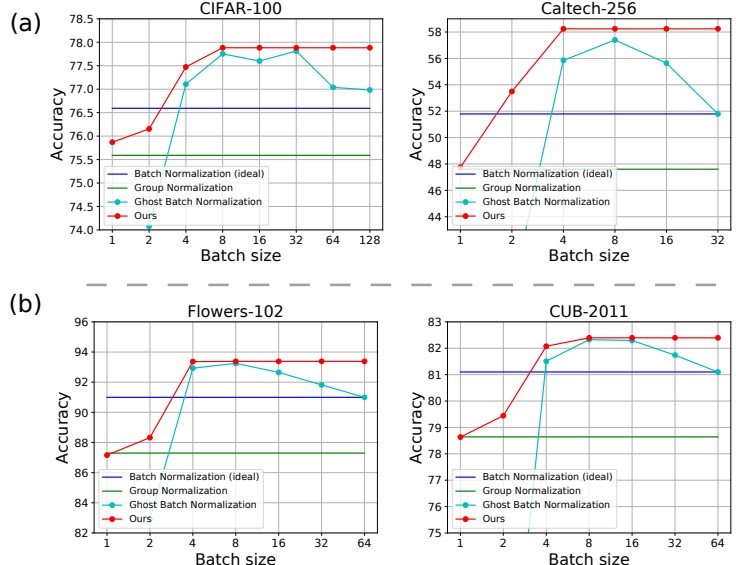

Figure 5: Total performance changes across batch sizes for CIFAR-100 and Caltech-256 (a) when training from scratch, incorporating all proposed improvements to Batch Normalization. On the bottom (b) is the same on Flowers-102 and CUB-2011, which employs transfer learning via fine-tuning from ImageNet. Also shown within each plot is the performance of Group Normalization, an idealized Batch Normalization that scales perfectly across batch sizes, and Ghost Batch Normalization (Sec. 3.2) by itself, for which the x-axis represents the Ghost Batch Size $B'$.

are due to the introduced weight decay, and for $B = 2$ the generalization of Batch and Group Normalization leads to the improvement (Sec. 3.4), with some additional effect from weight decay.

Improvements on Caltech-256 follow the same trends, but to greater magnitude, with a total increase in performance of 6.5% over Batch Normalization and an increase of 5.9% over Group Normalization for $B = 2$.

### 4.3 Transfer Learning

We also show the applicability of these approaches in the context of transfer learning, which we demonstrate on the Flowers-102 (Nilsback & Zisserman, 2008) and CUB-2011 (Wah et al., 2011) datasets via fine-tuning a ResNet-152 model from ImageNet. These tasks presents several challenges: 1) the Flowers-102 data only contains 10 images per category in the training set (and CUB-2011 only 30 examples per class), 2) pre-training models on ImageNet is a very strong form of prior knowledge, and despite the small dataset size may heavily reduce the regularization effects of some of the techniques, and 3) we examine the setting of pre-training with generic ImageNet models trained *without* any of these modifications, which gives an advantage to both the generic Batch Normalization and Group Normalization, for which pre-trained models exist.

We plot results in Fig. 5 (b), where we find remarkable qualitative agreement of our non-transfer learning results to this setting, despite the challenges. In total, on Flowers-102 our techniques were able to improve upon Batch Normalization by 2.4% (from 91.0% to 93.4% top-1 accuracy, a 27% relative reduction in error), and upon Group Normalization by 6.1% (from 87.3%, a 48% relative reduction in error). On CUB-2011, which has more training data, we improved upon Batch Normalization by 1.4% (from 81.1% to 82.4%) and Group Normalization by 3.8% (from 78.6%).

We anticipate that even further improvements might arise by additionally pre-training models with some of these techniques (particularly Ghost Batch Normalization), as we were able to see a large impact (roughly 5%) on Group Normalization by pre-training with a Group Normalization-based model instead of Batch Normalization.

Table 1: Accuracy on CIFAR-100 with non-i.i.d. minibatches. $B'$ refers to the Ghost Batch Normalization size (equivalent to the batch size for Batch Normalization and Batch Renormalization), and "Batch Group Norm." refers to our approach in Sec. 3.4. "Inf. Ex. Weight: Off" refers to using only the moving averages for normalization statistics (*i.e.* $\alpha = 0$), while "On" refers to tuning $\alpha$ based on the validation set.

| Method | $B'$ | Inf. Ex. Weight: Off | Inf. Ex. Weight: On |
|---|---|---|---|
| Batch Norm | 128 | 40.1 | 62.2 |
| Batch ReNorm | 128 | 69.0 | 69.0 |
| Ghost Batch Norm | 64 | 42.3 | 50.8 |
| | 32 | 57.8 | 70.9 |
| | 16 | 64.3 | 72.2 |
| | 8 | 68.7 | 72.0 |
| | 4 | 70.4 | 71.5 |
| | 2 | 68.4 | 71.4 |
| Batch Group Norm. | 2 | 75.2 | 76.1 |

## 4.4 Non-i.i.d. minibatches

An implicit assumption in Batch Normalization is that training examples are sampled independently, so that minibatch normalization statistics all follow roughly the same distribution and training statistics are faithfully represented in the moving averages. However, in applications where training batches are *not* sampled i.i.d., such as metric learning (Oh Song et al., 2016; Movshovitz-Attias et al., 2017) or hard negative mining (Shrivastava et al., 2016), violating this assumption may lead to undesired consequences in the model. Here, we test our approaches in this challenging setting.

Following Batch Renormalization (Ioffe, 2017), we study the case where examples in a minibatch are sampled from a small number of classes — specifically, we consider CIFAR-100, and study the extreme case where each minibatch ($B = 128$) is comprised of examples from only four random categories (sampled with replacement), each of which is represented with 32 examples in the minibatch. We present results for Batch Normalization, Batch Renormalization, our generalization of Batch and Group Normalization from Sec. 3.4 ("Batch Group Norm."), and the full interaction of Ghost Batch Normalization and Inference Example Weighing in Table 1. In this challenging setting, Inference Example Weighing, Ghost Batch Normalization, and Batch Group Norm all have large effect, in many cases halving the error rate of Batch Normalization. For example, Inference Example Weighing was able to reduce the error rate by 20% without any retraining, and tuning Ghost Batch Normalization, even without any inference modifications, was just as effective as Batch Renormalization, a technique partially designed for the non-i.i.d. case. Even further, Batch Group Normalization was hardly affected at all by the non-i.i.d. training distribution (76.1 vs 76.2 for i.i.d.). Last, it is interesting to note that Inference Example Weighing had practically no effect on Batch Renormalization (improvement $\leq 0.1\%$), confirming Batch Renormalization's effect in making models more robust to the use of training vs moving average normalization statistics.

## 5 Conclusion

In this work, we have demonstrated four improvements to Batch Normalization that should be known by all who use it. These include: a method for leveraging the statistics of inference examples more effectively in normalization statistics, fixing a discrepancy between training and inference with Batch Normalization; demonstrating the surprisingly powerful effect of Ghost Batch Normalization for improving generalization of models without requiring very large batch sizes; investigating the previously unstudied effect of weight decay on the scaling and shifting parameters $\gamma$ and $\beta$; and introducing a new approach for normalization in the small batch setting, generalizing and leveraging the strengths of both Batch and Group Normalization. In each case, we have done our best to not only demonstrate the effect of the method, but also provide guidance and evidence for precisely which cases in which it may be effective, which we hope will aid in their applicability.

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

## A   PROOF OF BATCH NORMALIZATION OUTPUT BOUNDS

Here we present a proof of Eq. 2. We first prove the bound as an inequality and then show that it is tight. Without loss of generality, we assume that $x_0$ is the minimum of $\{x_i\}_{i=0}^{B-1}$ and that $\gamma \geq 0$. Then we want to show that

$$\min_{x_0,\ldots,x_{B-1}} \gamma \frac{x_0 - \mu_i}{\sqrt{\sigma_i^2 + \epsilon}} + \beta = -\gamma\sqrt{B-1} + \beta \tag{5}$$

Expanding $\mu_i$ and $\sigma_i^2$ (using the maximum likelihood estimator for $\sigma_i^2$), and canceling the scaling and offset terms $\gamma$ and $\beta$, we want to show

$$\min_{x_0,\ldots,x_{B-1}} \frac{x_0 - \frac{1}{B}\sum_{i=0}^{B-1} x_i}{\sqrt{\frac{1}{B}\sum_{i=0}^{B-1}(x_i - \frac{1}{B}\sum_{j=0}^{B-1} x_j)^2 + \epsilon}} = -\sqrt{B-1} \tag{6}$$

From here we assume without loss of generality that $x_0 = 0$ – since the output of Batch Normalization is invariant to an additive constant on all $x_i$, we can subtract $x_0$ from all $x_i$ and maintain the same value. We also assume that all $x_i \geq 0$, then frame the minimum as a bound

$$\frac{-\frac{1}{B}\sum_{i=0}^{B-1} x_i}{\sqrt{\frac{1}{B}\sum_{i=0}^{B-1}(x_i - \frac{1}{B}\sum_{j=0}^{B-1} x_j)^2 + \epsilon}} \geq -\sqrt{B-1} \tag{7}$$

$$-\frac{1}{B}\sum_{i=0}^{B-1} x_i \geq -\sqrt{\frac{B-1}{B}\sum_{i=0}^{B-1}\left(x_i - \frac{1}{B}\sum_{j=0}^{B-1} x_j\right)^2 + \epsilon} \tag{8}$$

$$-\frac{1}{B}\sum_{i=0}^{B-1} x_i \geq -\sqrt{\frac{B-1}{B}\sum_{i=0}^{B-1}\left(x_i^2 - \frac{2x_i}{B}\sum_{j=0}^{B-1} x_j + \frac{1}{B^2}\left(\sum_{j=0}^{B-1} x_j\right)^2\right) + \epsilon} \tag{9}$$

$$\sum_{i=0}^{B-1} x_i \leq \sqrt{\sum_{i=0}^{B-1}\left((B-1)Bx_i^2 - 2(B-1)x_i\sum_{j=0}^{B-1} x_j + \frac{B-1}{B}\left(\sum_{j=0}^{B-1} x_j\right)^2\right) + \epsilon} \tag{10}$$

$$\sum_{i=0}^{B-1} x_i \leq \sqrt{(B-1)B\sum_{i=0}^{B-1} x_i^2 - 2(B-1)\left(\sum_{i=0}^{B-1} x_i\right)^2 + (B-1)\left(\sum_{i=0}^{B-1} x_i\right)^2 + \epsilon} \tag{11}$$

$$\sum_{i=0}^{B-1} x_i \leq \sqrt{(B-1)B\sum_{i=0}^{B-1} x_i^2 - (B-1)\left(\sum_{i=0}^{B-1} x_i\right)^2 + \epsilon} \tag{12}$$

$$\left(\sum_{i=0}^{B-1} x_i\right)^2 \leq (B-1)B\sum_{i=0}^{B-1} x_i^2 - (B-1)\left(\sum_{i=0}^{B-1} x_i\right)^2 + \epsilon \tag{13}$$

$$B\left(\sum_{i=0}^{B-1} x_i\right)^2 \leq (B-1)B\sum_{i=0}^{B-1} x_i^2 + \epsilon \tag{14}$$

$$\left(\sum_{i=0}^{B-1} x_i\right)^2 \leq (B-1)\sum_{i=0}^{B-1} x_i^2 + \epsilon \tag{15}$$

Using the fact that $x_0 = 0$ and $\epsilon > 0$, it suffices to show

$$\left(\sum_{i=1}^{B-1} x_i\right)^2 \leq (B-1)\sum_{i=1}^{B-1} x_i^2 \tag{16}$$

With a change of variables, we have the more general

$$\left(\sum_{i=0}^{N-1} x_i\right)^2 \le N \sum_{i=0}^{N-1} x_i^2 \tag{17}$$

$$\frac{1}{N^2}\left(\sum_{i=0}^{N-1} x_i\right)^2 \le \frac{1}{N} \sum_{i=0}^{N-1} x_i^2 \tag{18}$$

$$\mathbb{E}[x]^2 \le \mathbb{E}[x^2] \tag{19}$$

$$\mathbb{E}[x^2] - \mathbb{E}[x]^2 \ge 0 \tag{20}$$

which is simply an alternate form for the variance of x, which is always non-negative, completing the bound.

To show that the bound is tight, we can set $x_0 = 0$ and $x_i = a$ for all $i > 0$, where $a$ is a non-negative constant:

$$\frac{-\frac{1}{B}\sum_{i=0}^{B-1} x_i}{\sqrt{\frac{1}{B}\sum_{i=0}^{B-1}(x_i - \frac{1}{B}\sum_{j=0}^{B-1} x_j)^2 + \epsilon}} \tag{21}$$

$$\frac{-\frac{1}{B}\sum_{i=1}^{B-1} a}{\sqrt{\frac{1}{B}\left(\frac{(B-1)^2}{B^2}a^2 + \sum_{i=1}^{B-1}(a - \frac{B-1}{B}a)^2\right) + \epsilon}} \tag{22}$$

$$\frac{-\frac{B-1}{B}a}{\sqrt{\frac{1}{B}\left(\frac{(B-1)^2}{B^2}a^2 + (B-1)a^2\left(1 - \frac{2(B-1)}{B} + \frac{(B-1)^2}{B^2}\right)\right) + \epsilon}} \tag{23}$$

$$\frac{-(B-1)a}{B\sqrt{\frac{a^2(B-1)}{B}\left(\frac{B-1}{B^2} + 1 - 2\frac{B-1}{B} + \frac{(B-1)^2}{B^2}\right) + \epsilon}} \tag{24}$$

$$\frac{-(B-1)a}{\sqrt{a^2(B-1)\left(\frac{B-1}{B} + B - 2B + 2 + \frac{(B-1)^2}{B}\right) + \epsilon}} \tag{25}$$

$$\frac{-(B-1)a}{\sqrt{a^2(B-1)\left(\frac{B^2-2B+1+B-1-B^2+2B}{B}\right) + \epsilon}} \tag{26}$$

$$\frac{-(B-1)a}{\sqrt{a^2(B-1) + \epsilon}} \tag{27}$$

As $a \to \infty$ (or if $\epsilon = 0$), then this approaches

$$\frac{-(B-1)a}{a\sqrt{(B-1)}} \tag{28}$$

which is simply

$$-\sqrt{(B-1)} \tag{29}$$

completing the proof.

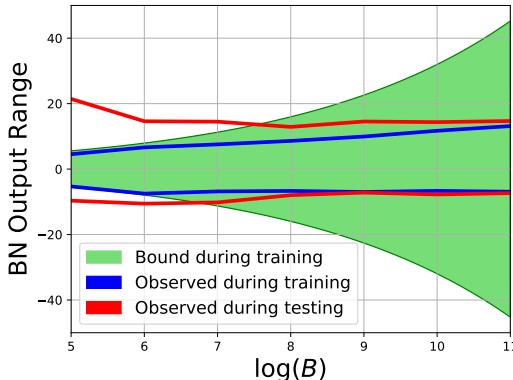

Figure 6: Range of output values obtained during inference on the CIFAR-10 test set, compared with the range observed during training and the bound of Eq. 2. See text for details.

## B    EMPIRICAL EVIDENCE OF BATCH NORMALIZATION OUTPUT BOUNDS

In Fig. 6 we show the observed output ranges for the last Batch Normalization layer in our CIFAR-10 network (spatial resolution: $4 \times 4$), plotting both the range during training and at inference time on the CIFAR-10 test set. Different values of $B$ were obtained by using different Ghost Batch Normalization sizes, keeping in mind that $B$ is determined by the product of the batch size and spatial dimensions.

At large values of $B$, it is unlikely that any network obtains a value even close to the bound of Eq. 2, but as $B$ gets smaller, the output range of the network during training becomes smaller in magnitude, eventually being nearly tight with our bound — for example, for $\log(B) = 5$, the theoretical minimum is $-5.57$, while the network obtained a minimum of $-5.30$. However, the maximum and minimum values obtained during inference on the test set show no clear pattern as $B$ changes, and are not subject to the training time bound, which is particularly noticeable for small values of $B$, where values fall outside the training-time bounds.

## C    NEGATIVE RESULTS: APPROACHES THAT DIDN'T WORK.

Here we detail a handful of approaches which seemed intuitively promising but ultimately failed to produce positive results.

BATCH NORMALIZATION MOVING AVERAGES.    In an attempt to resolve the other disparities Batch Normalization has between its training and inference behaviors, we experimented with a handful of different approaches for modifying the moving averages used during inference. First, since examples at inference time do not have data augmentation applied to them, we tried computing the moving averages over examples without data augmentation (implemented by training the model for a few extra epochs over non-augmented examples with a learning rate of 0, but still updating the moving average variables). This decreased accuracy on CIFAR-100 by roughly half a percent, though it did yield mild improvements to the test set cross-entropy loss.

Next, we experimented with calculating the moving averages over the *test set*, not making use of any of the test labels. Perhaps surprisingly, this behaved very similar to when moving averages were calculated over the training examples (within $0.1\%$ in accuracy and within $1\%$ in cross-entropy), with trends holding regardless of whether data augmentation was applied or not.

ADDING BATCH NORMALIZATION-LIKE STOCHASTICITY TO GROUP NORMALIZATION.    One of the hypotheses for why Group Normalization generally performs slightly worse than Batch Normalization is the regularization effect of Batch Normalization due to random minibatches producing variability in the normalization statistics. Therefore, we tried introducing stochasticity to Group Normalization in a variety of ways, none of which we could get to work well: 1) Adding gaussian

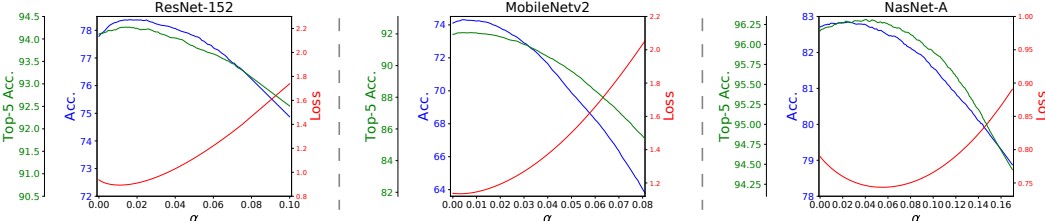

Figure 7: Effect of the example-weighing hyperparameter $\alpha$ on ImageNet; supplemental version of Fig. 1 with a larger range of $\alpha$.

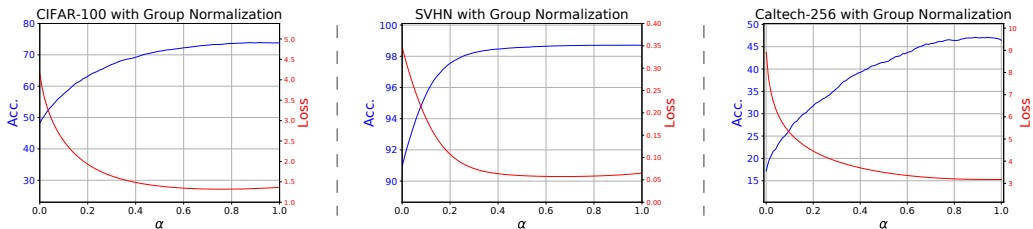

Figure 8: Effect of the example-weighing hyperparameter $\alpha$ for models trained with Group Normalization on CIFAR-100, SVHN, and Caltech-256; supplemental version of Fig. 2 with a larger range of $\alpha$.

noise to the normalization statistics, where the noise is based on a moving average of the normalization statistics, 2) Using random groupings of channels for calculating normalization statistics (optionally only doing randomization a fraction of the time), and 3) changing the number of groups throughout the training procedure, either as increasing or decreasing functions of training steps.

MORE PRINCIPLED GROUP SIZE COMPUTATION.   As part of generalizing Batch and Group Normalization, we examined whether it was possible to determine the number of groups in each normalization layer in a more principled way that simply specifying it as a constant throughout the network. For example, one approach we had mild success with was setting the number of elements per group (height × width × group size) to a constant, making the number of elements contributing to the normalization statistics uniform across layers. However, we were unable to get any of these ideas to work in a way that generalized properly across datasets. We also tried learning group sizes in a differentiable way with Switchable Normalization, but found that this made models overfit too much.

## D   SUPPLEMENTAL INFERENCE EXAMPLE WEIGHING PLOTS

In Figures 7, 8, and 9 we present plots corresponding to Figures 1, 2, and 4 of the main text, with larger ranges of the inference weight $\alpha$. In the main text, we restricted the range of $\alpha$ to values which showed off the tradeoff of $\alpha$ versus performance at a reasonably local scale, and these figures show a larger scale for completeness in characterizing model behavior. While this behavior can largely be extrapolated from the behavior for a smaller range of $\alpha$, there are some interesting trends.

On ImageNet 7, we see that only a small amount of inference example weighing is necessary to get most of its benefit, and setting $\alpha$ to larger values corresponds to a regime quite different than in training, smoothly decaying model performance as $\alpha$ becomes less and less appropriate. Similarly, when applying inference example weighing to Group Normalization (Fig. 8, while performance intuitively decays as $\alpha$ moves farther and farther away from 1, a surprisingly large range of values for $\alpha$ result in similar performance to Group Normalization, especially on SVHN. Lastly, when comparing the effect of $\alpha$ on models trained with Ghost Batch Normalization (Fig. 9, we clearly see that the optimal value for $\alpha$ is decreasing with respect to the Ghost Batch Normalization size, with the possible unusual exception of optimizing for loss on SVHN.

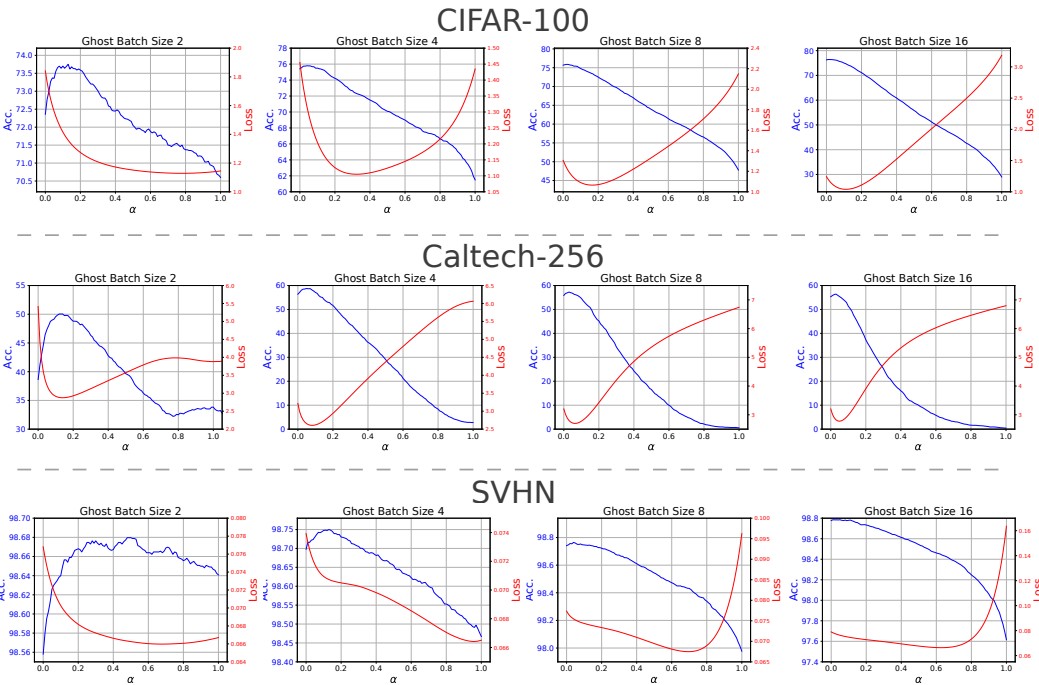

Figure 9: The complementary effects of Inference Example Weighing and Ghost Batch Normalization on CIFAR-100, SVHN, and Caltech-256; supplemental version of Fig. 4 with a larger range of $\alpha$.

