# OpenReview forum: "Four Things Everyone Should Know to Improve Batch Normalization"
_ICLR.cc/2020/Conference — Accept (Poster)_

### Official Review · AnonReviewer1 · 2019-10-23
**Official Blind Review #1**

**Rating:** 3

**Review:**

The authors discuss four techniques to improve Batch Normalization, including inference example weighing, medium batch size, weight decay, the combination of batch and group normalization. Equipped with the proposed techniques, the authors obtain promising results when training deep models with various batch sizes. However, the novelty of this paper seems very limited and more experiments are required.

Please see my detailed comments below.

Positive points：
1. The proposed inference example weighing method yields promising results and does not require any re-training.
2. The combination of batch and group normalization makes it possible to train deep models with very small batch size.
3. By combining all the techniques, the proposed method yields promising performance when training deep models with different batch sizes.

Negative points:

1. Some notations are very confusing. For example, the authors use B to represent the size of a minibatch. However, why do the authors only consider B-1 samples in Eq. (2), i.e., selecting the minimum possible output among x_1, …, x_{B-1}?

2. The proposed inference example weighing method seems very similar to Batch Renormalization. Both methods seek to use a linear function to combine the batch statistics and the moving statistics. What is the essential difference between these methods?

3. What model do the authors use in the experiment of Figure 2? Why do the authors conduct experiments on different datasets in Section 3.1 (including ImageNet) and Section 3.2 (excluding ImageNet)? It would be stronger to provide ImageNet results in Section 3.2.

4. The authors draw different conclusions about the usage of weight decay from a recent work (He et al, CVPR2019). The CVPR paper reports that training \gamma and \beta without weight decay on ResNet-50 yields significant performance improvement. However, this paper shows that training ResNet-50 with weight decay improves the performance. Please comment on the differences in the conclusions.

Reference: "Bag of tricks for image classification with convolutional neural networks." CVPR, 2019.

5. The authors only report ImageNet results of the proposed inference example weighing method. However, all the experiments in Section 4 are performed on three small datasets. It is necessary and important to provide ImageNet results to show the effectiveness of the other three techniques in Section 4.

6. Note that training deep models with non-i.i.d. minibatches is a typical case to evaluate normalization methods, e.g., Batch Renormalization. Specifically, examples in a minibatch are not sampled independently. What would happen if the authors apply the proposed techniques to the non-i.i.d. case?

7. Some closely related work should be discussed in the paper, such as

[1] "Decorrelated Batch Normalization." CVPR, 2018.
[2] "Double Forward Propagation for Memorized Batch Normalization." AAAI, 2018.
[3] "Differentiable Dynamic Normalization for Learning Deep Representation." ICML, 2019.
[4] "Iterative Normalization: Beyond Standardization towards Efficient Whitening." CVPR, 2019.

Minor issues:
1. In Section 1, the third contribution is not a complete sentence.

2. There are many typos in the paper.
(1) In Section 2, “Layer Normalization, which has found use in many natural language processing tasks.” Should “which has found use” be “which has been used”?
(2) In Section 3.1, “Batch Normalization has a disparity in function between training inference”. “between training inference” should be “between training and inference”.
(3) In Section 3.1, “we need only figure out …” should be “we only need to figure out …”



**Experience Assessment:**

I have published one or two papers in this area.

**Review Assessment: Checking Correctness Of Derivations And Theory:**

I carefully checked the derivations and theory.

**Review Assessment: Checking Correctness Of Experiments:**

I carefully checked the experiments.

**Review Assessment: Thoroughness In Paper Reading:**

I read the paper thoroughly.

---

> ### Author Response · Authors · 2019-11-15
> **Thank you for your review (part 2)**
>
> (continued from part 1)
>
> 7. Thank you for the references, we have added brief descriptions of each to our section detailing related work.
>
> Minor issues:
> With all due respect, only one of these is actually a typo (missing the word 'and' for 2.2):
>
> 1. The contributions are written as a list (note that each ends in a comma). Since contribution 4 is the last item in the list, contribution 3 ends with "and".
> 2.1. This is correct English grammar.
> 2.2. Thank you! This was indeed a typo.
> 2.3. This is correct English grammar.
>
> Thank you for the review!
> Best,
> Paper 1113 Authors

---

> ### Author Response · Authors · 2019-11-15
> **Thank you for your review (part 1)**
>
> (Split into two responses due to character limit)
>
> Thank you for your review! We appreciate that you valued the contributions of the paper (positive points 1-3).
>
> We aim to address your remaining concerns here:
>
> Novelty: We believe that our accumulation of insights is a significant contribution and of interest to the research community. Batch Normalization is fundamental to the success of much of deep learning. We presented multiple (four) improvements to it, validated our results extensively on six datasets in multiple training scenarios (training from scratch, transfer learning, and non-i.i.d. training in our updated version), and even presented intriguing negative results (see Appendix).
>
> Experiments: In the submitted version, we validated our methods on 5 datasets (with ImageNet the only dataset that didn't receive the whole gamut of experiments due to computational constraints). The other reviewers explicitly highlighted the experiments ("thorough empirical study", "support from experiments", "experimental evidence seems sufficient"). In our revision, we have added a 6th dataset, adding further evidence to the transfer learning setting, and also added the suggested experiments in the non-i.i.d. setting.
>
> 1. We wrote Eq. 2 as minimizing over only $x_1, \ldots, x_{B-1}$ because the minimum is actually independent of $x_0$ -- in other words, no matter what value $x_0$ has, we can pick $x_1, \ldots, x_{B-1}$ large enough that the bound is achieved (see proof in Appendix). Nonetheless, we appreciate that this might not be straightforward to see, and have revised the minimization to include $x_0$, with added explanation in the Appendix proof.
>
> 2. While both Inference Example Weighing and Batch Renormalization involve interactions with the moving average statistics, they're actually quite different:
> -Batch Renormalization applies a scaling and offset to make the offset and scaling used during training more similar to the moving average; Inference Example Weighing uses a weighted average of the example (not batch) statistics and moving averages during inference.
> -Batch Renormalization requires re-training a network; Inference Example Weighing doesn't.
> -Batch Renormalization introduces 2 hyperparameters, $r_{max}$ and $d_{max}$, and up to 5 in its full form when using a warmup as suggested in their paper (number of steps to use vanilla Batch Normalization, number of warmup steps for $r_{max}$, and a distinct number of warmup steps for $d_{max}$); Inference Example Weighing has a single hyperparameter that can be easily tuned, even on networks that have already been trained.
> -At its core, Inference Example Weighing applies during inference and doesn't affect training; Batch Renormalization applies during training and doesn't affect inference (other than indirect effects on weights).
>
> 3a. The choice of model was described in Sec. 4.1 (with a forward pointer in the first paragraph of Sec. 3): On CIFAR-100 and SVHN we used a ResNet-18, and on Caltech-256 we used Inception-v3.
> 3b. While we admire the work and interesting results of those who have the resources to do so, given our computational budget it was prohibitive to do many experiments on ImageNet (besides those in Sec. 3.1) -- a single model training on ImageNet takes about as long as all of the experiments put together on another dataset. Given that hyperparameters on ImageNet might also need to be tuned, we thought it more important to focus on fully exploring experiments on other datasets.
>
> Also, to demonstrate the real-world applicability of our methods, we performed transfer learning experiments, which are directly applicable to most real-world scenarios; they employ transfer from a large, labeled dataset (ImageNet) in order to solve problems when less data is available, as is typically the scenario for all use-cases where labeled data isn't plentiful. We have also strengthened our transfer learning experiments in the revision by adding an additional transfer learning dataset.
>
> 4. Interestingly, while He et al. discuss removing regularization on \gamma and \beta, which they combine with any other biases or variables in the network, in their experiments (Table 4 of their paper) removing this weight decay actually led to a degradation in performance (note that each row of the table stacks the previous rows' changes together). Their experiments are consistent with our results.
>
> 5. (See 3b, as the question is very similar.)
>
> 6. Although this wasn't the primary goal of our paper, this is an interesting experiment, which we have performed and now added to our paper. In short, Inference Example Weighing and Ghost Batch Normalization both help tremendously in the non-i.i.d. setting (up to ~30% reduction in error) by reducing the dependence on the distribution within each batch of data. This shows that our techniques are beneficial in both i.i.d. and non-i.i.d. settings. Thank you for your suggestion!

---

### Official Review · AnonReviewer2 · 2019-10-23
**Official Blind Review #2**

**Rating:** 6

**Review:**

The paper introduces four techniques to improve the deep network model through modifying Batch Normalization (BN). The inspirations are from the gaps between train&test and between batches in multi-gpu training, comparison to other normalization methods, and weight decay in regularizing convolution weights training. The paper studies each techniques with the support from experiments. The paper is easy to follow. The techniques seem effective.

The paper mentions "theory" multiple times, but lacks sufficient justification to support these "theories". So one suggestion is to replace "theory" with a soft word.

Experimental evidence seems sufficient and there are some theoretical derivations, but it looks incremental that the paper presents some techniques in improving Batch Normalization only. In general, the paper is of values to the community.

**Experience Assessment:**

I do not know much about this area.

**Review Assessment: Checking Correctness Of Derivations And Theory:**

I assessed the sensibility of the derivations and theory.

**Review Assessment: Checking Correctness Of Experiments:**

I assessed the sensibility of the experiments.

**Review Assessment: Thoroughness In Paper Reading:**

I made a quick assessment of this paper.

---

> ### Author Response · Authors · 2019-11-15
> **Thank you for your review**
>
> Thank you for your review! We put care into making sure the concepts in the paper were easy to follow, and appreciate that it came through effectively.
>
> Theory: While we do have some theory in our paper (e.g. the proof in Appendix A), we have replaced the instances of this word with "hypothesis", which is more clear for the other parts of the paper. Thank you for your suggestion!
>
> Novelty (same comment as Reviewer 3): We appreciate and agree with the reviewer's opinion that improving Batch Normalization, such a fundamental component of modern neural networks, is of significant value to the community. We presented multiple (four) improvements to Batch Normalization, validated our results extensively, and in our latest revision we have added experiments on another transfer learning dataset, bringing the number of evaluation datasets up to 6, and added an additional experiment for training on non-i.i.d. distribution.
>
> Thank you for the review!
> Best,
> Paper 1113 Authors

---

### Official Review · AnonReviewer3 · 2019-10-23
**Official Blind Review #3**

**Rating:** 6

**Review:**

The paper performs an empirical study of four batch-normalization improvements and proposes a new normalization technique for small batch sizes, based on group and batch normalizations. Among others, the authors address the inconsistency between the train and the test stages and the problem of small batch sizes. The authors conducted an empirical ablation study of the four techniques and proposed an intuition when each method should be used.

Concerns:
(1) The comparison with baselines in section 4.2 seems to be unclear. Fig.5 shows the performance of normalization methods for different batch sizes. Batch Normalization, however, has the same performance for all batch sizes. The authors refer to this baseline as “idealized Batch Normalization”. Additional elaboration on what does this means is required.
(2) It would also be beneficial to see the comparison with the original Ghost Batch Normalization in the final evaluation (section 4.2), since this method, according to section 3.2, was capable of the significant improvement for Caltech-256 dataset.
(3) In section 3.1, the authors provide an intuition of why can the discrepancy between test and train phases hurts the performance of a model. The empirical evaluation of this effect is needed to justify this intuition.

Overall, the newly proposed method is a minor update, and novelty is limited. However, the thorough empirical study of existing improvement techniques would be a good addition to the conference.

Minor comments:
1. share the y-axis in Fig.4 between different ghost batch sizes.

I would also recommend authors to include the following papers to the related work section:
1. Riemannian approach to batch normalization [https://arxiv.org/abs/1709.09603]

----------

Respond to the rebuttal. Clarification on the concern (3):

I agree that, in general, a discrepancy between training and testing can hurt a model. The paper showed that the output of a batch normalization layer is theoretically unbounded during testing. However, it would be beneficial to see numerically if it indeed the case on a real test set (the output range is wider than the one during training).



**Experience Assessment:**

I have published in this field for several years.

**Review Assessment: Checking Correctness Of Derivations And Theory:**

I assessed the sensibility of the derivations and theory.

**Review Assessment: Checking Correctness Of Experiments:**

I assessed the sensibility of the experiments.

**Review Assessment: Thoroughness In Paper Reading:**

I read the paper thoroughly.

---

> ### Author Response · Authors · 2019-11-15
> **Thank you for your review**
>
> Thank you for the thoughtful review! We appreciate your thorough reading of the paper, and here are responses to your listed concerns:
>
> (1) This was meant to provide an easy comparison to Batch Normalization -- in the ideal case, Batch Normalization would scale perfectly across batch sizes, not changing in performance as the overall batch size changed. Plotting the performance of Batch Normalization this way lets us compare our approach against that ideal.
>
> (2) Thank you for the suggestion, we have now added standalone Ghost Batch Normalization to the analysis in Figure 5 of the updated paper.
>
> (3) We're a little unsure of the precise meaning of what you're asking, and would appreciate any clarification you could give. Generally speaking, introducing most types of discrepancies between training and testing should hurt a model, since models perform best when they are trained to directly optimize the task they are evaluated on. While some exceptions exist (e.g. data augmentation), training on a task or distribution slightly different from the task being evaluated on, or training in a slightly different way than the evaluation settings, should hurt performance. Resolving discrepancies introduced by Batch Normalization has been a common theme in prior work (see references in Sec. 3.1)
>
> Novelty (same comment as Reviewer 2): We appreciate and agree with the reviewer's opinion that improving Batch Normalization, such a fundamental component of modern neural networks, is of significant value to the community. We presented multiple (four) improvements to Batch Normalization, validated our results extensively, and in our latest revision we have added experiments on another transfer learning dataset, bringing the number of evaluation datasets up to 6, and added an additional experiment for training on non-i.i.d. distribution.
>
> Fig. 4 y-axis: Thank you for your suggestion! We've looked into this, but found that keeping the y-axis constant across different plots makes many of the plots hard to read -- for example, for Caltech-256 the full y-axis for accuracy would be [30, 60], but Ghost Batch Sizes 4 through 16 only have values in the range [53, 60], which squeezes all of the accuracy curves down to ~25% of their original size.
>
> Reference: Thank you for the reference! We have added it in our related work.
>
> Thank you for the review!
> Best,
> Paper 1113 Authors

---

> ### Author Response · Authors · 2019-12-23
> **Response to Clarification (3)**
>
> (We wanted to add this response earlier but it was outside the paper rebuttal period)
>
> Thank you for the clarification -- yes, in addition to the theory, this is also empirically true, and we can provide multiple concrete examples of networks exceeding our bound in Eq. 2 during testing, but respecting it during training, where the bound applies. We have measured this on seven networks, and all seven exhibited a wider output range during testing than during training. We will add these experiments to the Appendix.

---

### Public Comment · ~Micah_Goldblum1 · 2019-11-08
**An Interesting Connection**

Hi Authors,
Thank you for your interesting paper.  I noticed that your results concerning weight decay for networks with batch normalization are related to our paper [1], which examines alternatives for influencing effective learning rate, especially in networks with batch norm.  Please consider mentioning the relationship with our work in your next version.

[1] https://arxiv.org/abs/1910.00359

---

> ### Author Response · Authors · 2019-11-15
> **Thank you for the reference**
>
> Yes, always happy to cite the latest and greatest research (this paper was submitted to arxiv after the ICLR submission deadline).
> Thanks for the reference!

---

### Public Comment · ~Saurabh_Singh1 · 2019-12-20
**Prior work on example weighing in BN**

Dear authors,

Kudos on acceptance. Please note that your observations regarding the example weighing have previously been made and studied in the following paper as well.

EvalNorm: Estimating Batch Normalization Statistics for Evaluation
https://arxiv.org/abs/1904.06031

Your eq. 3 is quite similar to eq. 3 and 4. in the above paper (though with a slight difference)

Quite uncannily, the above work focuses on evaluation time adjustment of normalization statistics as you say in your paper "An advantage of this technique is that we can apply it retroactively to any model trained with Batch Normalization, allowing us to verify its efficacy on a wide variety of models. " and makes precisely this point.

I would encourage the authors to include a discussion and comment on differences in the relevant section in their next revision.

---

> ### Author Response · Authors · 2019-12-23
> **Response**
>
> Hi Saurabh,
>
> Thank you for the reference, and kudos as well on having your paper accepted to ICCV.
>
> There indeed is similarity between the method you present in your paper and one of the four methods we study ("Inference Example Weighing", Section 3.1). For what it's worth, we posted an earlier version of this work on arxiv over six months ago, well before the referenced paper was published in ICCV, and had actually performed initial experiments with the method several months before that.
>
> From what we can tell, there are a number of differences between the method we propose and the one studied in your paper:
> -'EvalNorm' uses different weighing parameters for the mean and variance, additionally using different parameters in each normalization layer (other than for a few baselines studied). Our approach uses a single parameter for both moments and all layers.
> -'EvalNorm' optimizes its weighing parameters by encouraging similarity to the training outputs of a BN layer; we treat our single weighing parameter $\alpha$ as a hyperparameter to be optimized for a metric of interest -- indeed, as we demonstrate in Figures 1, 2, and (especially) 4, we show that optimizing for different metrics, even if they're correlated, can result in rather different optimal values of $\alpha$.
> -We give a proof on the effect of Inference Example Weighing on the output range of a Batch Normalization layer.
> -We examine more thoroughly how model behavior changes with the weighing parameter.
> -The experimental setups and evidence are quite different -- for example, we also study the non-i.i.d. minibatch setting and show the impact of Inference Example Weighing, while the referenced paper also looks at object detection.
>
> Overall, we view 'EvalNorm' and Inference Example Weighing from our work as taking two complementary approaches towards addressing the same limitation of Batch Normalization, where both were clearly done in parallel, a relatively common occurrence when studying important problems.
>
> We'd be happy to cite your work, and should you find yourself updating the arxiv version of your paper, would appreciate if you do the same.

---

### Decision · Program_Chairs · 2019-12-19

**Decision:**

Accept (Poster)

**Comment:**

This paper proposes techniques to improve training with batch normalization. The paper establishes the benefits of these techniques experimentally using ablation studies. The reviewers found the results to be promising and of interest to the community. However, this paper is borderline due in part due to the writing (notation issues) and because it does not discuss related work enough. We encourage the authors to properly address these issues before the camera ready.